# Molecular insights into high-altitude adaption and acclimatisation of *Aporrectodea caliginosa*

Iain Perry[1,2] , Szabolcs Balazs Hernadi[1], Luis Cunha[3,4] , Stephen Short[1,5], Angela Marchbank[6], David J Spurgeon[5] , Pablo Orozco-terWengel[1] , Peter Kille[1]

Here, we explore the high-altitude adaptions and acclimatisation of *Aporrectodea caliginosa*. Population diversity is assessed through mitochondrial barcoding, identifying closely related populations across the island of Pico (Azores). We present the first megabase N50 assembly size (1.2 Mbp) genome for *A. caliginosa*. High- and low-altitude populations were exposed experimentally to a range of oxygen and temperature conditions, simulating altitudinal conditions, and the transcriptomic responses explored. SNP densities are assessed to identify signatures of selective pressure and their link to differentially expressed genes. The high-altitude *A. caliginosa* population had lower differential expression and fewer co-expressed genes between conditions, indicating a more condition-refined epigenetic response. Genes identified as under adaptive pressure through $F_{st}$ and nucleotide diversity in the high-altitude population clustered around the differentially expressed an upstream environmental response control gene, HMGB1. The high-altitude population of *A. caliginosa* indicated adaption and acclimatisation to high-altitude conditions and suggested resilience to extreme weather events. *This mechanistic* understanding could help offer a strategy in further identifying other species capable of maintaining soil fertility in extreme environments.

## Introduction

Earthworms provide a critical role as terrestrial eco-engineers in the recycling of biological material, improving soil fertility, and are commonly used as part of environmental management to improve poor soils (Butt, 2010; Blouin et al, 2013). The higher temperatures and increased extreme weather events associated with global warming has increased the frequency of crop losses (Powell & Reinhard, 2016). This has led to some crops being increasing planted at higher altitudes where cooler conditions can be found, elevating the importance of understanding mountain soil fertility (Lenoir et al, 2008; Skarbø & VanderMolen, 2015). Significant investment has been focused on improving the resilience of cultivated crops, as well as developing strategies to manage exposure to extreme weather events at low land and facilitate crop cultivation on mountainsides (Gale, 2004; Zorn et al, 2005; Butt & Briones, 2011; Skarbø & VanderMolen, 2015; Tolessa et al, 2017). Earthworms play a critical role in decomposition, nutrient cycling, and crop yield; the impact of climate change on earthworm communities has been recently comprehensively been reviewed (Singh et al, 2019). However, this review has highlighted the limited consideration that has been afforded to understanding the resilience of annelid species at high altitudes that are critical for soil fertility, resilience vital to their persistence at such locations.

High-altitude mountains at temperate latitudes present extreme environmental conditions with large temperature fluctuations and changes in oxygen availability as atmospheric pressures decrease with altitude (Peacock, 1998). Although vagile species may cope with such environmental stressors by changing their location, less mobile species must adapt or acclimatise to those conditions to avoid perishing. As ectotherms earthworms are vulnerable to temperature changes (Costanzo & Lee, 2013), and although some anecic species use deep burrows to avoid temperature extremes, shallow burrowing endogeic and epigeic species are particularly susceptible. Adaption and acclimatisation of humans and other mammals to high altitude, and in particular hypoxia, has been an area of significant research (Manalo et al, 2005; Ke & Costa, 2006; Majmundar et al, 2010; Scheinfeldt & Tishkoff, 2010; Storz et al, 2010), but few studies have attempted to investigate altitudinal impacts on earthworms (Gonzalez et al, 2007; Kanchilakshmi, 2016) Earthworms rely upon a relatively unique system that uses cutaneous respiration and erythrocruorin to deliver oxygen to its tissues (Giardina et al, 1975; Royer et al, 2006), and mechanisms of response to atmospheric oxygen deprivation for this system are not well described.

To investigate adaptive and acclimatised responses of earthworms to high altitude, we can look at differential expression of genes and at variation in the distribution of single-nucleotide polymorphisms (SNPs) across the genome that may be indicative of selective processes (Robledo et al, 2017). However, although such

[1]Organisms and Environment, Cardiff University, Wales, UK  [2]Wales Gene Park, Cardiff University, Wales, UK  [3]Department of Life Sciences, Centre for Functional Ecology, University of Coimbra, Coimbra, Portugal  [4]School of Applied Sciences, University of South Wales, Wales, UK  [5]UK Centre for Ecology and Hydrology, Maclean Building, Wallingford, UK  [6]Genomics Hub, Cardiff University, Wales, UK

Correspondence: perryia3@cardiff.ac.uk

analyses are readily feasible for many species that have had their genomes assembled, earthworms, as a group, are underrepresented among these with *Eisenia fetida* (N50 < 10 Kb), *Eisenia andrei* (N50 ~ 740 Kb), *Metaphire vulgaris* (N50 ~ 4.2 Mb), and *Amynthas corticis* (N50 ~ 31 Mb), representing the few earthworms with their genomes sequenced (Zwarycz et al, 2015; Bhambri et al, 2017 Preprint; Jin et al, 2020; Shao et al, 2020; Wang et al, 2021). Assembling a high-quality genome (N50 > 10 Kbp) for an earthworm has proved a challenge for many years because of their high allelic diversity (Rimington, 2022); however, with long-read technology combined with scaffolding techniques, it is possible to now overcome this hurdle (Ghurye & Pop, 2019).

A fortuitous combination of geography and human migration offers us the chance to investigate molecular mechanisms underlying the capacity of earthworms to adapt to high altitudes. The geologically young Azores archipelagos began emerging a little over 8 million years ago, but the youngest island. Pico, volcanically grew only around 270 1,000 yr ago (Miranda et al, 1998; Carine & Schaefer, 2009) with the main volcanic peak of Pico erupting only 300 yr ago (Woodhall, 1974). Estimates of human settlement with farming range from ca. 15[th] century to potentially as far back as the 11[th] century (Pacheco et al, 2010; Rodrigues et al, 2015). It is likely that many of the species, including earthworms, on the island are semi-invasive, brought over with human travel from multiple locations (Novo et al, 2015). Even with the earliest of potential earthworm stowaways, there has been very little time for species to adapt to the island and less time still to adapt to high altitude after eruptions.

In this study, we develop the first genome for *Aporrectodea caliginosa* and generate a masked and annotated genome. We then go on to use this new resource in conjunction with differential expression, $F_{st}$ and nucleotide diversity (Pi) to help to identify the mechanisms by which *A. caliginosa* has adapted or acclimatised to extreme climate conditions found at high altitudes as derived from populations resident to low and high altitudes in the Island of Pico within the Azorian archipelago.

# Results

## Species and population selection

Of the 165 *A. caliginosa* individuals collected only for COII analysis (Fig 1A), for 619 nucleotide sites, Theta-W was 0.00427, Pi 0.00562, and Tajima's D 0.6265 ($P$ > 0.10). These suggest a low level of genetic diversity across the island of Pico and when combined with Baeyesian skyline analysis, indicate that the seeding population had undergone a long-term population contraction since the last glacial maximum (Fig 1B).

The origins of Pico *A. caliginosa* was assessed through generation of haplotype groups and calculating a time rooted phylogeny compared with Western Palearctic *A. caliginosa* individuals (Fig 1C). A distinct linage split between northern Spain and the French Pyrenees *A. caliginosa* individuals was detected, diverging between 27.9 and 107.4 million years ago. However, individuals from Pico clustered into multiple haplotypes found in several areas across the Western Palearctic including Ireland, France, Austria, and Southern Spain, potentially indicating multiple introduction events.

## Genome generation

More than 19.5 μg high–molecular-weight (HMW) pure DNA was extracted from a single *A. caliginosa* individual from Pico (2,200 m asl). Sequencing generated a total of: 24.59 Gbp of long-read (~10 Kbp) nanopore reads (from three runs), 97 Gbp of 10X chromium–barcoded PE150 reads, and 60 Gbp of PE150 short-read data. An initial assembly of paired-end error corrected nanopore reads with Wtdbg2 generated an N50 of 159 Kbp. This was successively increased to an N50 of 1.2 Mbp with scaffolding from 10X chromium reads using NanoChrome. BUSCO indicated 93.4% of core metazoan genes are present. Genome annotation identified 42,566 gene objects across the masked genome, relating to 25,556 genes were identified (Fig 2).

## Differential expression analysis

Between 4 and 11 million reads were sequenced for each individual, with mapping statistics to the genome similar between both high- and low-altitude population individuals. When all samples were analysed together through DESeq2, and principal component analysis (PCA) run, principal component 2 split the expression profile of native individuals with experimental individuals, whereas principal component 3 clearly separates HA and LA populations (Fig 3A).

The LA population had a larger number of genes showing differential expression between treatment pairs than the HA population. Differentially expressed genes for each population in each condition were assessed via DiVenn. This indicated that many of the differentially expressed genes between treatments for the LA population overlapped indicating a shared response, whereas the HA population indicated a more nuanced response with a reduced level of differential expression between conditions (Fig 3B–D). Of particular note, the environmental response high-mobility group (HMG) genes, *HMGB1* and *HMGB2*, were universally up-regulated with a *P*adj < 0.05 in both HA and LA populations in all temperature comparisons of 4°C versus 21°C (with fold changes for both populations ranging between 1.8 and 3.2 for *HMGB1* and 1.8 and 3.3 for *HMGB2*). Exposure to different temperatures appears to have the greater impact on differential gene expression than a reduction in atmospheric oxygen.

## SNP analysis

A phylogeny of individuals assessed in the differential gene expression analysis and SNP analysis was calculated from their SNPs across the genome. There was a clear difference in SNP compliments between HA and LA individuals, indicating a rapid accumulation of SNPs separating the HA population from the LA population (Fig 4A). The LA native individuals taken from a separate LA native fauna site indicate a slight divergence from those collected for LA experimental exposures.

The PCA and MDS plots (Fig 4B and C) generated from the SNPs support the divergence observed in this phylogeny, whereas the Manhattan plot indicates the number of genomic regions under selective pressure (Fig 4D). A total of 358 genes passed filters for $F_{st}$ and nucleotide diversity (Pi) (Fig 4E). A further filter was applied to

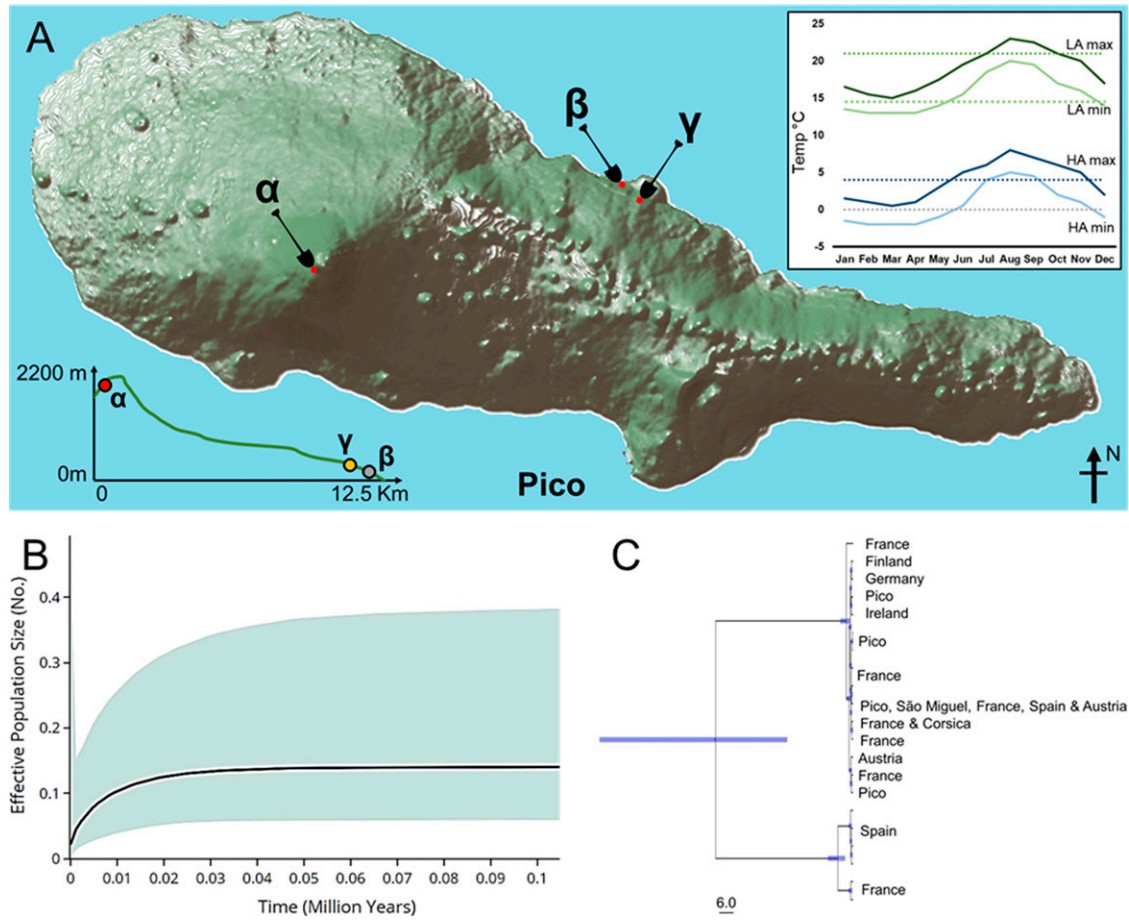

**Figure 1.  Population geography and history.**
**(A)** Approximate location and altitude of dig sites for island and high-altitude (HA) experimental and native expression individuals (α), low-altitude (LA) experimental individuals (β), and LA native expression individuals (γ). **(B)** Long-term population decline of *Aporrectodea caliginosa* started during the last glacial maximum. Bayesian skyline plot analysis of COII sequences drawn from individuals from across the island (Table S1) shows a median value and ranges between upper and lower confidence interval shaded in blue. **(C)** Dated phylogeny estimating divergence time for *A. caliginosa*, calculated using COII haplotype groups with trees generated on BEAST v1.8.4. and visualised with TreeAnnotator v2.4.8. The time scale is in 6 million years, and blue node bars indicate the confidence interval of branch splits.

remove genes with fewer than five SNPs per gene leaving 271 genes identified as under selective pressure.

To identify if any of the 271 genes had direct interactions with each other or to the genes *HMGB1* and *HMGB2* that were identified as universally up-regulated in HA and LA differential expression, interactions were examined in STRING (Fig 5A). This analysis identified that although *HMGB1* is not itself under selective pressure, its interaction with *TP53* forms the epicentre of a network of genes that are. Through putative transcription factor enrichment analysis and upstream expression to kinase analysis, upstream control of SNP-associated genes were identified in differentially expressed genes (Fig 5B), and the overlap between all sets of identified genes are identified (Fig 5C) with *SOX2* appearing in all four gene lists.

## Discussion

To explore the adaptions and acclimatisation of earthworms to altitude, a population of *A. caliginosa* was identified on the strato-volcanic island of Pico, Azores. The population introduced to the island demonstrates the historical bottlenecking of diversity from the last glacial maximum, but no further expansion or contraction is detected. The date of introduction of *A. caliginosa* to the island has not yet been identified; however, the evidence of genetic bottlenecking suggests this is no older than 10,000 yr ago, and more probably occurred with the arrival of human settlers with transplanted crops in the last millennium (Ashe, 1813; Pacheco et al, 2010; Rodrigues et al, 2015). All measures of mitochondrial diversity indicated a low genetic diversity between high- and low-altitude populations. Critically, this suggested that populations could be used effectively and be operated close to the model inbred mouse strains that are used in mammalian research.

The genome generated as a tool for investigating differential expression and SNPs is the first genome for the species *A. caliginosa* and is one of the first megabase assemblies for earthworms. This presents a significant advance in the resources for one of the most cosmopolitan and widely distributed earthworm species that is used heavily in environmental monitoring and for research (Bart et al, 2018). The use of NanoChrome software greatly improved the assembly of scaffolds, with some contigs now close to

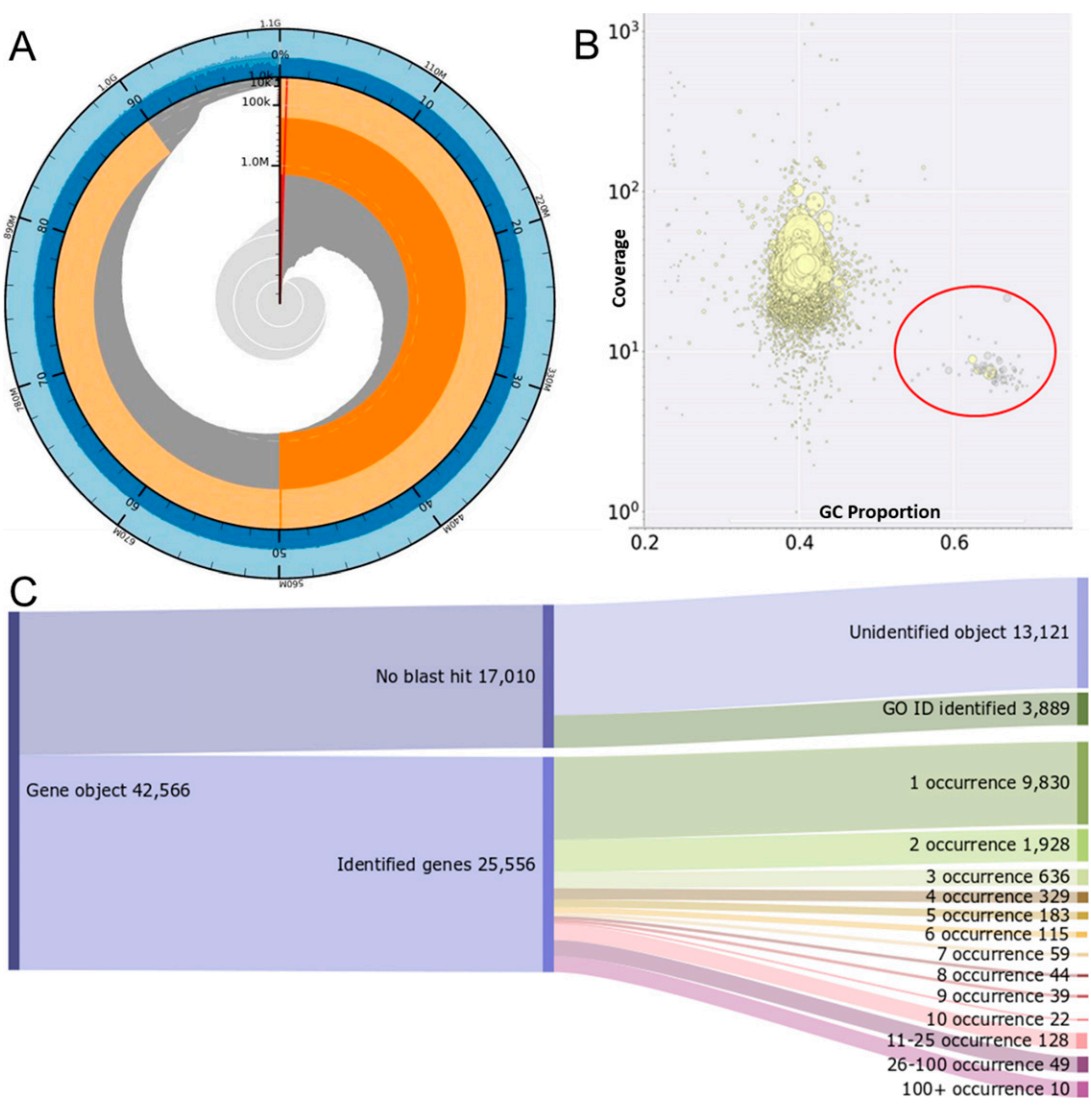

**Figure 2. Genome assembly and annotation statistics.**
**(A)** Assembly statistics of the assembled *Aporrectodea caliginosa* genome. Genome size is 1.1 Gbp, longest contig is 6.5 Mbp and N50 is 1.2 Mbp. A total of 279 contigs account for 50% of the genome assembly. A total of 530 contigs account for 90% of the genome. GC composition is 40.2%. **(B)** Identification of bacterial contamination including verminephrobacter (red circle) with BlobTools before removal from genome assembly. **(C)** Identification of genome annotations and the number of genes with multiple occurrences within the genome reveals most of the identified genes occur only once.

chromosome arm length. The assembly demonstrated high completeness as assessed via BUSCO and purity of assembly as assessed via BlobTools. The megabase assembly is one of the most contiguous assemblies for an earthworm assembly by a considerable size and leave a short step to chromosome level assembly. The future use of chromosome conformation capture (Hi-C) should allow the final scaffolding of a full diploid chromosomal assembly from the scaffolds present, overcoming the very high levels of haploid diversity.

In replicating high-altitude conditions, exposures to low temperatures (having spent 6 mo at 21°C) induced more differential gene expression than reduction in oxygen concentrations. This is not unexpected as the cutaneous respiration of earthworms and their use of erythrocruorin avoids the well-explored pressures of

respiration in mammals (Giardina et al, 1975; Royer et al, 2006; Storz, 2016). With increasing adverse weather events, earthworms may be exposed to lowered levels of oxygen on a more regular basis, for example, fluctuations in soil saturation from heavy rain that can drive worms from burrows to surface. For many populations, these conditions are temporary; however, earthworms have been found to survive in other extreme environments with very low levels of oxygen and high levels of carbon dioxide (volcanic caldera) (Cunha et al, 2011). Similarly, for those identified at high altitude, there is no temporary escape to a more oxygen rich surface. Despite the acclimatisation of LA and HA earthworm populations in identical conditions for 6 mo, a timeframe that should have reduced interference from native differential expression of genes that are not determined by hereditary factor, the LA population still had a

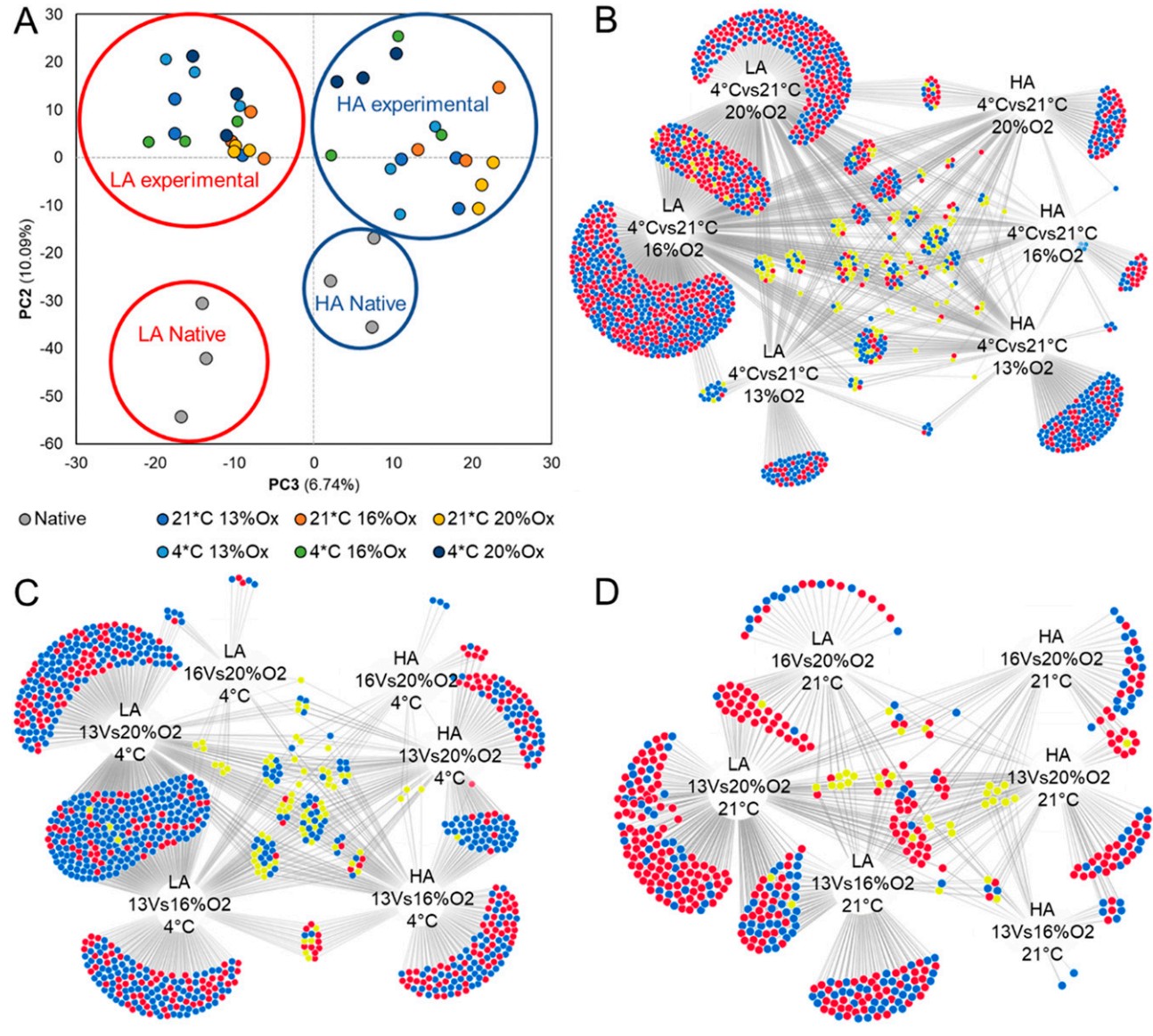

**Figure 3. Differential gene expression analysis.**
**(A)** Principal component analysis plot of all experimental and native individuals. High-altitude (HA) individuals (blue circle) separate cleanly from the low-altitude (LA) individuals (red circle) via PC3. **(B, C, D)** Identification of shared differentially regulated genes for both HA and LA individuals (up-regulate red, down-regulated blue, conflicting yellow). **(B, C, D)** Conditions investigated: (B) temperature at three oxygen concentrations, (C) Oxygen concentrations at 4°C and (D) oxygen concentrations at 21°C.

significantly higher level of differentially regulated genes to both temperature and oxygen availability than the HA population. As indicated in PCA analysis, where native expression of HA and LA populations group with experimental HA and LA populations, response to the simulated climatic conditions suggests that populations' responses are rooting from either adaptive changes or from epigenetics that prime for rapid environmental response beyond the 6 mo preincubation. In the HA population, fewer genes show differential expression between paired temperature comparisons, potentially indicating a more nuanced condition specific response to survival in extreme weather. The HA population's adaption through transcriptome remodelling could be because their existing physiological systems were more adaptable. The high

levels of differential expression in the LA population suggest a widespread adaptive change founded in a stress-like response. Thus, in the LA population, a number of common stress response pathways such as ERK/MAPK and Wnt signalling were differential expressed, indicating a rapid modulation of protective stress pathways that are not required in the HA population. The identification of *HMGB1* and *HMGB2* as universally up-regulated is of particular note. The primary function of these two genes is in the regulation of environmental response, operating as a DNA chaperone for pathways involved in (but not limited to) inflammation, immune response, DNA repair, and hypoxia response (Muller et al, 2004; Kang et al, 2014). These genes act as chromatin-binding factors or, through extracellular release, binding to RAGE and

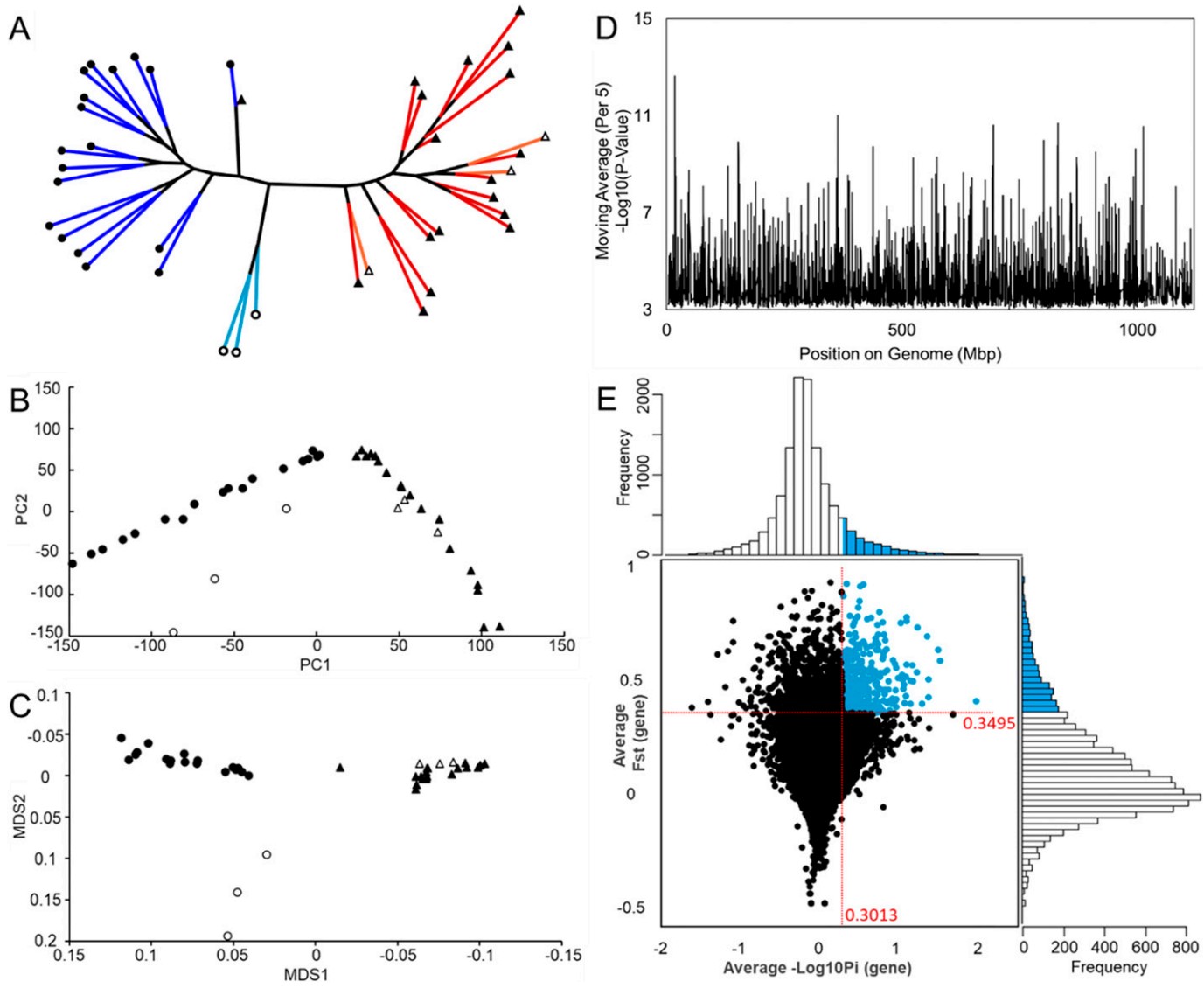

**Figure 4. Single-nucleotide polymorphism (SNP) analysis from transcriptomic reads.**
**(A)** SNP phylogeny separation of high-altitude (HA) individuals (experimental red, native orange) from low-altitude (LA) individuals (experimental blue, native cyan). LA natives taken from a separate low altitude site are slightly diverged from LA experimental individuals. **(B)** Principal component analysis separation of HA and LA individuals. **(C)** MDS separation of HA and LA individuals. **(D)** SNP *P*-value distribution across the genome as moving average indicating areas of elevated diversity. **(E)** Filtering distribution of SNPs with average FST per gene and average nucleotide diversity per gene.

TLRs and activating MAPK and NFκB (Bianchi, 2009). Despite both populations displaying up-regulation of *HMGB1* and *HMGB2* in response to the temperature shift, the lower levels of differentially expressed genes within the HA population suggests that the HA population might be using a separate response mechanism not used in the LA population.

SNP analysis identified a large number of genes as under selective pressure in the HA population compared with the LA population. Many of these genes are responsible for proteins that form interacting networks that HMGB1 critically forms the epicentre of. That network involves multiple transcription factors and cell cycle regulators such as NFκB1 which was identified as under selective pressure and directly interacts with HMGB1 itself has a

wide variety of biological interactions, particularly in stress response (Concetti & Wilson, 2018). *HMGB1*'s role in acclimatory thermal acclimatisation has been previously explored in detail including their role as "general activators" rather than solely specific gene regulators (Somero, 2005). *HMGB1* functions to allow increased gene-specific transcription factors. Although in the LA population this explains the high levels of differential gene expression, the pattern is not repeated in the HA population. It is possible that the lower levels of differential gene expression seen in the HA population is because of some or a combination of the changes identified in the networks of interacting genes, changes to upstream promoters outside of coding regions sequenced in this study, or epigenetic modification. These changes minimise energy

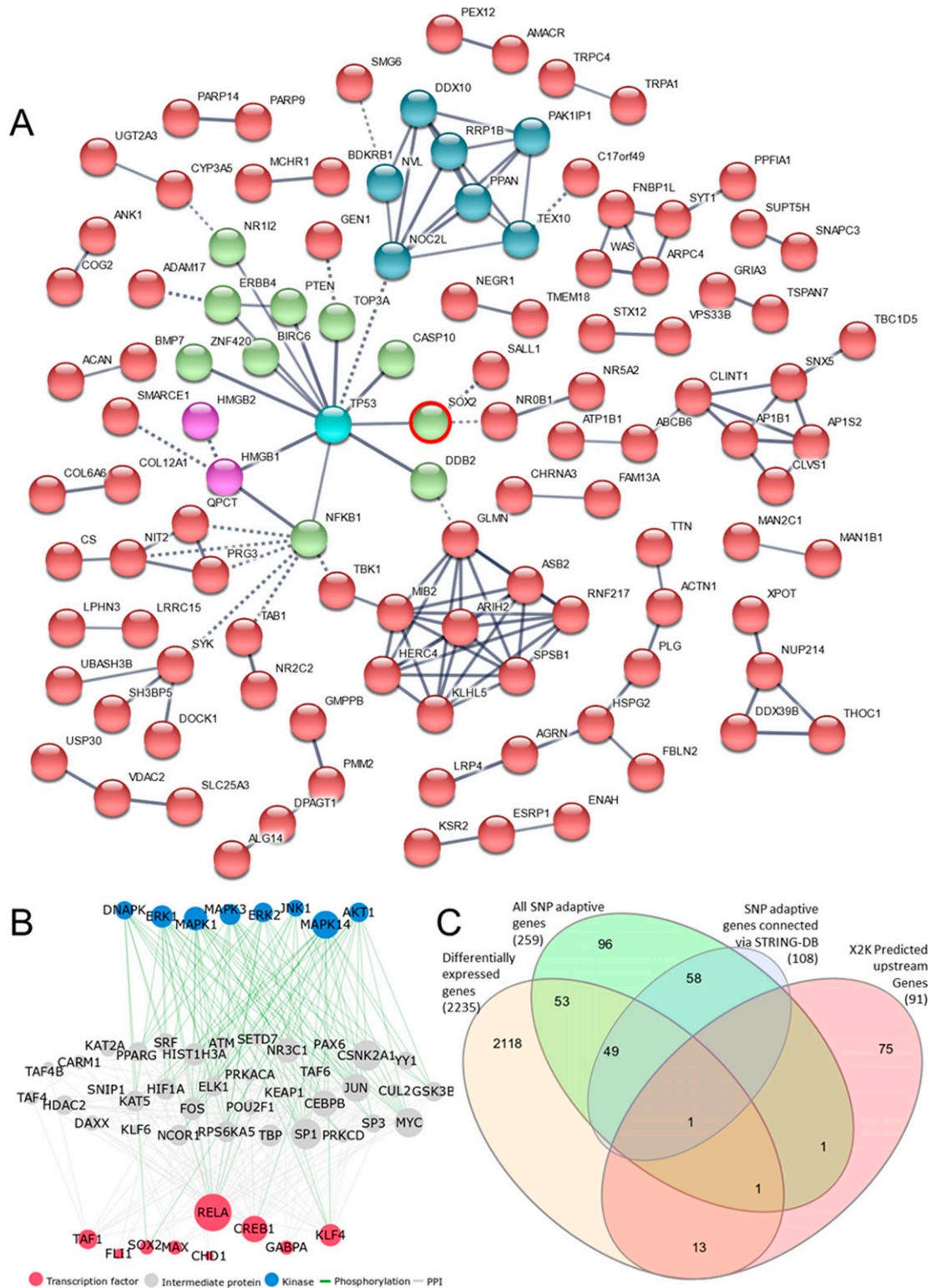

**Figure 5. Genes with high number of single-nucleotide polymorphisms have direct interactions with the differentially regulated genes.**
**(A)** Kmeans clustering of *HMGB1*, *HMGB2* (pink), and *TP53* (cyan) interacting with genes under selective pressure determined by STRING-DB (STRING genes) with a high-confidence (0.7) minimum interaction score. *HMB1*, *HMGB2*, *TP53*, and cluster 1 genes (green) are responsible for "DNA topological change"; cluster 2 genes (blue) are responsible for "ribosome biogenesis"; and *TBK1*, *GLMN*, *MB2*, *ASB2*, *ARIH2*, *RNF217*, *SPSB1*, *HERC4*, and *KLHL5* within cluster 3 (red) are responsible for "protein ubiquitination." **(B)** X2K interaction of kinases, TFs and upstream activation pathway to STRING genes. **(C)** Overlaps between differentially expressed genes, genes under selective pressure, STRING genes, and genes upstream of STRING identified by X2K.

expenditure to these stressors and are providing long-term adaption to the extreme weather shifts seen at high altitude.

The significant limitation of the current study is the absence of fitness data which prevent us from determining comparative phenotypic adaptation between the two populations. However, the genetic changes identified are indicative of adaption and acclimatisation of the HA population which is particularly re-markable considering the last volcanic eruption 300 yr ago is likely to have erased any existing populations, meaning the current HA population represents a LA population that has returned and rapidly adapted and acclimatised (Woodhall, 1974). Earthworms are not alone in rapid island evolution as mor-phological evolution has been observed in several families and taxa including mammals, lizards, and crabs (Schubart et al, 1998; Millien, 2006; Eloy de Amorim et al, 2017), though few studies have attempted to understand the genetics behind the observed morphological changes in even isolated mainland environments (Lyman et al, 2002).

## Conclusions

The population of *A. caliginosa* from Pico, Azores, indicates a putative epigenetic acclimatisation and adaption to environ-mental stressors. In particular, the HA population appears ca-pable of rapid acclimatisation to changes in extreme weather events without the need for transcriptional modification of large numbers of genes. This opens the prospect of their ability to survive increasingly common extreme weather events associated with human accelerated global warming climatic change. High-altitude environments could prove critical population either for reseeding resilient populations after localised extinction events or for the engineering of improved fertility HA soils for growing crops at HA. The identification of other species that are similarly endowed is an important area for further investigation.

# Materials and Methods

## Species and population selection

*A. caliginosa* individuals from across the island of Pico (Table S1), Azores (38.467N 28.400W), were collected in 96% EtOH. DNA ex-traction and purification were carried out with the QIAGEN DNeasy Blood and Tissue Kit (QIAGEN). Samples were quantified via NanoDrop (ND-1000) prior amplification of a fragment of the mi-tochondrial DNA COII gene (Pérez-Losada et al, 2009). The PCR amplification of the COII fragment consisted of an initial DNA denaturation at 95°C for 5 min followed by 36 cycles of (a) DNA denaturation at 95°C for 1 min, (b) annealing at 50°C, and (c) ex-tension of PCR products at 72°C for 1 min, ending with a final ex-tension at 72°C for 10 min. PCR reactions consisted of 25 $\mu$l reaction volumes containing: 5× PCR Buffer Flexi Green, 2.5 mM MgCl$_2$, 10 mM dNTPs, 1 U Taq polymerase (Promega), 10 pM COII primers (Pérez-Losada et al, 2009), and ~50 ng of template DNA. PCR products were visualised via Qiaxcel (QIAGEN) and sequenced by Eurofins Ge-nomics. Sequences were quality assessed before aligning and

trimming to the same length (619 bp) with MEGA (v7.0.26) (Kumar et al, 2016). Theta-W, Pi, and Tajima's D were calculated using DnaSP for the high and low populations separately (Rozas et al, 2017).

A time-calibrated Bayesian phylogenetic tree was calculated using BEAUti and BEAST (v1.8.4) with an estimated molecular di-vergence rate of 2.4% per site per million years (Chang & James, 2011). BEAST was run independently 10 times with the Markov Chain Monte Carlo algorithm run for 20 million steps and discarding the initial 25% of the steps as burn-in. After removing the burn-in the remaining of the runs were merged using LogCombiner. Conver-gence of the MCMC were assessed using Tracer (v1.6) (Drummond et al, 2012), with the later also used to estimate the haplotype network.

## HMW DNA isolation

Muscular tissue from a high-altitude (HA) individual from Pico was dissected (avoiding the clitellum and five segments from the tail tip) and digested for 6 h in 1,080 $\mu$l ATL buffer, 40 $\mu$l proteinase K at 20 mg/ml, and 20 $\mu$g/ml RNase (QIAGEN) at 56°C. DNA was purified via phenol/chloroform extraction and precipitated in isopropanol containing 0.2% ammonium acetate before hooking with a glass rod. The isolated DNA was then washed in 70% ethanol and dis-solved in elution buffer (QIAGEN) for 48 h. The resulting HMW DNA (fragments >40 Kbp) was assessed via NanoDrop, Qubit and agarose gel electrophoresis.

## Genome sequencing and assembly

Fifty-fold coverage of barcoded 10X chromium sequenced PE150 and 50-fold coverage of unbarcoded short-read sequencing PE150 was performed by Novogene. For nanopore sequencing, 5 $\mu$g of HMW DNA was fragmented to ~16 Kbp in a Covaris g-Tube before SPRI bead cleanup (Beckman-Coulter) to remove small-molecular-weight fragments below ~3,000 Kbp. The fragmented DNA was sequenced on three Oxford Nanopore MinION flow cells (MK1 R9 RevD) following manufacturer's protocols.

Combined nanopore long-read data were error-corrected using the short-read data with FMLRC (Wang et al, 2018) before assembly with Wtdbg2 (Ruan & Li, 2020). The assembly was further im-proved with transcriptomic data for *A. caliginosa* using L_RNA scaffolder (Xue et al, 2013). The large contigs were further scaffolded with three rounds of NanoChrome that uses barcoded 10x chromium reads supported by nanopore long-read data (Rimmington, 2019). Between each round, LR_Gapcloser and SOAP gap closing was run to fill gaps identified in NanoChrome scaffolding (Xu et al, 2018).

BUSCO v1.0, BlobPlots v1.0, and contig statistics were used to assess the quality of the assembly, and contaminating reads from the earthworm symbiote *Verminephrobacter* were removed (Laetsch & Blaxter, 2017; Waterhouse et al, 2017). Repeat masking of the genome assembly was performed with RepeatModeler and RepeatMasker (Smit et al, 2015). The masked assembly was used to generate an annotations file with OmicsBox, Blast2GO, and Inter-proscan (Biobam). Gene sequences in the masked genome were annotated by performing a BlastP search (E-value cutoff 10$^{-5}$) against the human proteome (UP000005640).

### Experimental setup, exposures, RNA extraction, library generation, and sequencing

Populations (n200 individuals) of earthworms were collected from Pico at high altitudes (HA) (38.46447N 28.40615W, 2,073 m asl) and low altitudes (LA) (38.49733N 28.26995W, 196 m asl). HA and LA "native" individuals were also collected in RNAlater (Thermo Fisher Scientific) for native RNA expression analysis. HA native individuals were collected from the same site as live HA individuals for experimental work. LA native individuals were taken from a LA site close (at a similar altitude) to the LA experimental individuals where endemic flora was present but where sufficient experimental LA worms could not be collected (38.48672N 28.25272W, 340 m asl).

Analysis of "native" individuals provides insight into impact of chronic exposure to the climatic difference that exists at high altitude. Experimental exposures were design probed the acute response indicative of short-term weather patterns. Before experimental exposures, HA and LA experimental populations were kept for 6 mo to acclimatise to the same conditions, that is, room temperature in 20 liters boxes of soil comprised of equal ratio of topsoil, compost, and bark chipping. Soil was kept moist, and earthworms were supplied with 0.5 Kg horse manure as feed every 2 mo. For the experimental study, earthworms were placed in identical containers (20 × 14 cm) with mesh lids in 1 kg of soil and exposed to various temperature and oxygen for 2 wk to replicate different climatic conditions. Conditions selected cover a factorial combination of high and low temperature and three oxygen levels, specifically, 21°C at 13% $O_2$, 21°C at 16% $O_2$, 21°C at 20% $O_2$ and 4°C at 13% $O_2$, 4°C at 16% $O_2$, 4°C at 20% $O_2$. HA and LA populations were exposed to each condition with 10 worms from each population per container. Worms were washed and weighed before and after exposure. Temperature was controlled in an Innova 4230 refrigerated incubator (New Brunswick Scientific) and oxygen with a ProOx model C2′ (BioShperix).

Immediately after the exposure, earthworms were hand-sorted from the soil, and RNA extracted from a 1-cm posterior section (five segments from tail tip, avoiding any clitellum tissue). Tissue was homogenised in 600 $\mu$l TRIzol (Thermo Fisher Scientific) before RNA was purified with the Direct-Zol RNA MiniPrep kit (Zymo Research). Purified samples were quantified by HS RNA Qubit (Invitrogen), and RNA quality assessed with a Qiaxcel RNA cartridge (QIAGEN). Three samples from the high- and low-altitude populations for each exposure condition were selected for sequencing, with individuals with the highest RNA concentration selected for the RNASeq library preparation. Libraries were generated with a KAPA mRNA Hyper-Prep kit (Roche). Samples were assessed via a D1000 chip on an Agilent Tapestation, and samples pooled evenly at 1 nM. All 42 samples were sequenced twice 1× 75 bp on a NextSeq 550 high-capacity chip.

### Differential expression analysis

Demultiplexed reads were quality trimmed with Trimmomatic (v2.8.3) (Bolger et al, 2014), duplicate reads with Piccard (Li et al, 2009), and mapped to the unmasked genome with STAR(v2.7) (Dobin et al, 2013). Differential read analysis between conditions was assessed with SARTools (v1.7.3) and DESeq2 (v1.12) (Varet et al, 2016).

Lists of identified differentially expressed genes were filtered with a $Log_2$fold cutoff of <−1.4 and >1.4 with an FDR $P_{adj}$ cutoff of <0.05. Differential gene expression patterns between conditions were assessed using GProfiler (e99_eg46_p14_f929183), DiVenn (v2.0), and STRING (v11.0) (Pomaznoy et al, 2018; Sun et al, 2019; Szklarczyk et al, 2019).

### SNP analysis

Trimmed RNASeq data used in differential expression analysis were mapped to the genome using the GATK best practices pipeline that uses STAR(v2.7), Piccard (v2.22.2), Samtools (v1.1), and GATK4 to generate a SNP VCF files for the HA and LA individuals (Li et al, 2009; McKenna et al, 2010; Dobin et al, 2013; Brouard et al, 2019; GATK-Team, 2019). Hard-quality filtering was performed on the VCF files to remove low coverage and poor-quality data (DP < 10, QD < 5, MQ < 40, SOR > 3). Using SNPhylo (v 20160204) (maximum percent of sample without SNP information [PNSS] 25 and max missing rate 0.5), a phylogenetic tree was calculated and visualised in FigTree (v1.4.3) (Lee et al, 2014; Rambaut, 2016). PCA and multidimensional scaling (MDS) were calculated with TASSEL (v5.2.6) (Bradbury et al, 2007).

General linear model analysis was run on all detected SNPs, and the $P$-values plotted as a Manhattan plot with TASSEL. $F_{st}$ per SNP (MAC 8) between HA and LA populations was calculated with VCFtools (v0.1.16). Tajima's Pi (per SNP) and Tajima's D (sliding window 30,000) were calculated for both HA and LA. Average $F_{st}$ and −$Log_{10}$ ratio of HA/LA Pi values per gene were calculated, and genes were subsequently filtered to retain the genes with the top 10% of values in both. These were further filtered to include genes where HA-Pi was less than or equal to half LA-Pi. Genes that had fewer than five SNPs were excluded. Genes that passed these filters were assessed with STRING to identify connections to differentially expressed genes identified in the differential gene expression analysis. Disconnected genes were hidden and a high-stringency filter used to mask low confidence gene–gene interactions. To identify putative transcription factors and upstream pathway interactions of genes identified in the STRING analysis, transcription factor enrichment analysis (TFEA) and expression-to-kinases analysis were performed with eXpression2Kinases (Clarke et al, 2018). Putative genes were identified in differentially expressed genes, and gProfiler gene enrichment performed.

## Data Availability

Data can be found at www.ncbi.nlm.nih.gov/bioproject/ under the project codes: PRJNA623866 – Genome assembly, PRJNA638118—low-altitude population transcriptome, PRJNA638117—high-altitude population transcriptome.

## Supplementary Information

**Life Science Alliance**

# Acknowledgements

Dr Stephen Short was supported by Natural Environment Research Council award number NE/R016429/1. Gratitude to Dr Paulino Sapo and the Serviço de Ambiente da Ilha do Pico for permission to research earthworms on the island.

## Author Contributions

I Perry: conceptualisation, data curation, formal analysis, investigation, visualization, methodology, project administration, and writing—original draft, review, and editing.
SB Hernadi: investigation and methodology.
L Cunha: investigation and methodology.
S Short: methodology and writing—original draft.
A Marchbank: resources.
D Spurgeon: writing—original draft.
P Orozco-terWengel: supervision, methodology, and writing—original draft.
P Kille: conceptualisation, data curation, supervision, funding acquisition, investigation, methodology, project administration, and writing—original draft, review, and editing.

## Conflict of Interest Statement

The authors declare that they have no conflict of interest.

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
