## [Reviewer comments · Life Science Alliance]

Life Science Alliance

Molecular insights into high altitude adaption and acclimatisation of *Aporrectodea caliginosa*

Iain Perry, Szabolcs Hernadi, Luis Cunha, Stephen Short, Angela Marchbank, David Spurgeon, Pablo Orozco-terWengel, and Peter Kille

DOI: <https://doi.org/10.26508/lsa.202201513>

Corresponding author(s): Iain Perry, Cardiff University

Review Timeline:

Submission Date:	2022-05-03
Editorial Decision:	2022-06-17
Revision Received:	2022-06-29
Editorial Decision:	2022-07-20
Revision Received:	2022-07-28
Accepted:	2022-07-29

Scientific Editor: Novella Guidi

Transaction Report:

June 17, 2022

Re: Life Science Alliance manuscript #LSA-2022-01513-T

Iain Perry

Dear Dr. Perry,

Thank you for submitting your manuscript entitled "High altitude adaptation and acclimatisation of *Aporrectodea caliginosa*: Molecular insights into a population resistant to extreme weather events at high altitude" to Life Science Alliance. The manuscript was assessed by expert reviewers, whose comments are appended to this letter. We invite you to submit a revised manuscript addressing the Reviewer comments.

Thank you for this interesting contribution to Life Science Alliance. We are looking forward to receiving your revised manuscript.

Sincerely,

B. MANUSCRIPT ORGANIZATION AND FORMATTING:

Reviewer #1 (Comments to the Authors (Required)):

The present paper provides experimental evidence to support acclimatization of earthworms to high elevation sites. The work is sound and the experimental procedure clear and convincing.

I have two observations.

First, the timeline of establishment of populations and thus the timeframe for acclimation could be a little more clear. I realise that some of these timings are only known in broad terms, but it is useful to know for example that the process of acclimation can happen pretty quickly. Also, this particular process could be just a little more connected to other studies on acclimation and adaptation. These are two separate processes that are likely to occur over different timeframes.

Second, while the acclimation potential is well documented, the relationship of that to extreme environments and climate change is pretty speculative. I think the connection to extreme environments and climate change is a useful point for discussion, and that discussion would connect to the findings presented, but is an emergent feature of the paper rather than a central premise. For example, how do the different environments relate to climate change? There is no evidence presented to connect the experimental conditions to climate change projections. In summary, there should be a clear delineation between the immediate results and further implications arising.

On the whole, I found this to be a sound piece of work that will be of broad interest.

Reviewer #2 (Comments to the Authors (Required)):

This paper presents some very good genomic data for an earthworm along with interesting expression data when two populations are exposed to different conditions which have relevance to climatic differences across an elevation gradient. However the paper is overinterpreted when it comes to adaptation and acclimation given the nature of the population sampling and conditions considered. It is great to see these types of analyses taking place on invertebrates but much more caution is required in making statements about adaptation and evolutionary processes when based on such information.

Note: it is annoying not to have line numbers on pages and even page numbers.

Abstract

I think the first few sentences overstate what is being done here. Genomics are a useful tool in understanding environmental adaptation but they are only a tool, they are not a replacement for the characterization of adaptive variation linked directly to fitness. We don't really know if populations are adapted just because they differ genomically. This is a problem throughout this paper. It is fine to do comparisons across populations and treatments but another matter to interpret these conclusively in terms of showing adaptation and acclimation. Be clear and circumspect about what is and is not being tested here. You have differences among two populations but without further work and replication it is difficult to make specific conclusions at this stage.

Introduction

Is there evidence that earthworms are changing in density in response to climate change? The arguments in the first couple of this section seem very indirect. Also I would be tempted to leave out mammals, there is plenty of relevant invertebrate data to relate to on elevation gradients. This section did not really get interesting to me until the bottom of the page.

Final paragraph needs restatement to indicate what this study is about. It is a shame that there is no population from the mainland.

Methods

It is an exaggeration to state that individuals were collected from across a range of elevations, there were effectively three sites and two were investigated in detail. The sampling of course is key here and the nature of the collections including numbers was not at all clear to me from the description given. This also applies to the sampling for the experimental analysis.

Was the earthworm symbiont present in all individuals? What fraction of reads was assigned to the symbiont? I assume that it was sequenced earlier though no reference is given here.

In the acclimation period, what was survival of earthworms like across the 6 months? Is there potential for selection here to change the genetic composition of the population? How was the exposure period for the worms determined? How did the number 42 arise for the analysis given there were 2 pop x 3 sample x 6 treatment combinations?

Results

COI data does not give a particularly good picture of genetic diversity as opposed to nuclear data or population history. How do

these compare?

The Fig 1 legend should indicate clearly the nature of the data referred to here in making B. these are very wide error bars and I wonder if this figure really indicates much particularly as the sampling across the island seems very limited.

In the absence of replication, it is hard to work out whether high and low elevation populations differ in the number of expressed genes. I found it hard to make sense of the high mobility group of genes initially as I read the paper. It is not really mentioned in the related Fig 3.

Was there an association between the length of the genes and the number of SNPs detected?

In the SNP analysis, I was unclear what the "rapid accumulation" comment referred to. Again difficult to determine in the absence of replicated populations at the different elevations.

The TP53 identification looked interesting though I am not familiar with this analysis and I wondered about how one controls for spurious network patterns - some additional explanation would have been useful.

Discussion

See above comments, this jumps too quickly into assumed adaptation and acclimation in the absence of fitness data and better replicated sampling without acknowledging the limitations of the current work.

Reviewer #1 (Comments to the Authors (Required)):

The present paper provides experimental evidence to support acclimatization of earthworms to high elevation sites. The work is sound and the experimental procedure clear and convincing.

I have two observations.

1. First, the timeline of establishment of populations and thus the timeframe for acclimation could be a little more clear. I realise that some of these timings are only known in broad terms, but it is useful to know for example that the process of acclimation can happen pretty quickly.

We believe paragraph 4 of the introduction sets out the timeframe for the populations with the arrival of humans to the islands and the eruption of the main volcanic peak where the high altitude worms were collected from (lines 88+)

2. Also, this particular process could be just a little more connected to other studies on acclimation and adaptation. These are two separate processes that are likely to occur over different timeframes.

The role of climate with other studies has been heavily covered by Singh et al. (Line 53) and while it does not specially compare adaption and acclimatisation timeframes, we feel this is a more comprehensive connection to other studies that do and avoids over-verbosity.

3. Second, while the acclimation potential is well documented, the relationship of that to extreme environments and climate change is pretty speculative. I think the connection to extreme environments and climate change is a useful point for discussion, and that discussion would connect to the findings presented, but is an emergent feature of the paper rather than a central premise. For example, how do the different environments relate to climate change? There is no evidence presented to connect the experimental conditions to climate change projections. In summary, there should be a clear delineation between the immediate results and further implications arising.

The limitations of this paper have been highlighted more transparently including absence of fitness data (line 292). The experimental conditions examined in the paper are replicative of current climates found at low and high altitudes. Clarified (in lines 384+) the experimental exposures for two weeks are indicative of rapid shifts in weather which is linked to climate change.

On the whole, I found this to be a sound piece of work that will be of broad interest.

Reviewer #2 (Comments to the Authors (Required)):

This paper presents some very good genomic data for an earthworm along with interesting expression data when two populations are exposed to different conditions which have relevance to climatic differences across an elevation gradient. However the paper is overinterpreted when it comes to adaptation and acclimation given the nature of the population sampling and conditions considered. It is great to see these types of analyses taking place on invertebrates but much more caution is required in making statements about adaptation and evolutionary processes when based

on such information.

1. Note: it is annoying not to have line numbers on pages and even page numbers.

Line numbers have now been included in this draft.

2. Abstract

I think the first few sentences overstate what is being done here. Genomics are a useful tool in understanding environmental adaptation but they are only a tool, they are not a replacement for the characterization of adaptive variation linked directly to fitness. We don't really know if populations are adapted just because they differ genomically. This is a problem throughout this paper. It is fine to do comparisons across populations and treatments but another matter to interpret these conclusively in terms of showing adaptation and acclimation. Be clear and circumspect about what is and is not being tested here. You have differences among two populations but without further work and replication it is difficult to make specific conclusions at this stage.

The abstract has been shortened to be in line with LSA length guidelines. This has removed the first few sentences and tempered some of the conclusions.

3. Introduction

Is there evidence that earthworms are changing in density in response to climate change? The arguments in the first couple of this section seem very indirect.

The role of earthworms has been highlighted, lines 51-53

4. Also I would be tempted to leave out mammals, there is plenty of relevant invertebrate data to relate to on elevation gradients. This section did not really get interesting to me until the bottom of the page.

Although as invertebrate biologist we appreciate the sentiment of the referee we feel the linkages to mammal studies enables accessibility to wider audience allow them to relate to the material presented. No action has been taken.

5. Final paragraph needs restatement to indicate what this study is about. It is a shame that there is no population from the mainland.

We have amended the final sentence of the introduction to define more specifically the study context. Where appropriate within the phylogenetic analysis we have related our island population to that of mainland haplotypes. We selected the Pico populations due to their low diversity seeming from bottle-necking when the islands were seeded. We did perform a feasibility study was performed at Les Deux Alps as a mainland comparison, however it was not possible to collect a *A. caliginosa* population in high enough numbers to replicate the experimental analysis.

6. Methods

It is an exaggeration to state that individuals were collected from across a range of elevations, there were effectively three sites and two were investigated in detail. The sampling of course is key here and the nature of the collections including numbers was not

at all clear to me from the description given. This also applies to the sampling for the experimental analysis.

We have refined the text to primarily reflect the three populations used within this specific study. The additional sampling sites have been included as supplemental (Table S1) and reflected in modification of Figure 1A. See Question 13.

7. Was the earthwork symbiont present in all individuals? What fraction of reads was assigned to the symbiont? I assume that it was sequenced earlier though no reference is given here.

Verminephrobacter presence was only specifically identified through blasting of genome assembly contigs where it formed 1 contig out of the 0.15% identified not annelid and subsequently removed for a contamination-free assembly.

8. In the acclimation period, what was survival of earthworms like across the 6 months?

Specific survival of earthworms was not assessed, however, sufficient numbers of individuals remained after the acclimation period for experimentation indicating along with no directly observed mortalities, earthworm survival was high.

9. Is there potential for selection here to change the genetic composition of the population?

If the reviewer is enquiring if the period of acclimation may have the potential for support selection for a sub-population we feel this is unlikely as significant mortality was not observed during this acclimation period.

10. How was the exposure period for the worms determined?

The following clarification has been added to the M&M section: Analysis of 'native' individuals provide insight into impact of chronic exposure to the climatic difference that exist at high altitude. Experimental exposures were design probed the acute response indicative of short-term weather patterns. (Lines 384-386)

11. How did the number 42 arise for the analysis given there were 2 pop x 3 sample x 6 treatment combinations?

2 pop v 3 sample of native (non-experimental) RNA expression for comparison.

12. Results

COI data does not give a particularly good picture of genetic diversity as opposed to nuclear data or population history. How do these compare?

COI data was used to confirm species and indicate that genetic diversity was comparatively low. The genome SNP based phylogeny (Fig 4A) calculated across the entire transcriptome describes in much greater detail, the genetic diversity.

13. The Fig 1 legend should indicate clearly the nature of the data referred to here in making B. these are very wide error bars and I wonder if this figure really indicates much particularly as the sampling across the island seems very limited.

We have added a clarification within the legend to identify the genetic data and individuals used for performing the analysis This cross references to Supplemental Table S1 which has been included to show the range of sites assessed (25 in total including the three used in RNAseq experimentation).

These sites contribute to the 165 worms detailed for generation of the island populations calculated diversity. The sites cover most of the island however the three sites alpha, beta and gamma were selected as the primary site collections due to species presence at high numbers.

14. In the absence of replication, it is hard to work out whether high and low elevation populations differ in the number of expressed genes. I found it hard to make sense of the high mobility group of genes initially as I read the paper. It is not really mentioned in the related Fig 3.

Replication is provided through the representation of multiple individuals drawn from HA and LA populations to derive those genes differentially expressed between the two populations. High-Mobility Group (HMG) genes were identified are differential expressed between population and during environment exposures as is clearly started (line $x - y$). Changes in expression of HMG genes in other HA and LA population may form the basis of future studies but are beyond the scope of the current investigation.

15. Was there an association between the length of the genes and the number of SNPs detected?

This parameter was not determined within the current analysis.

16. In the SNP analysis, I was unclear what the "rapid accumulation" comment referred to. Again difficult to determine in the absence of replicated populations at the different elevations.

SNPs frequencies within the two populations were determined from 18 individuals (all individuals used for transcriptomic analysis). The comment referenced by the referee relates to the intra-population frequency of SNPs within this significant group of individuals. The comment associated with the rapidity of accumulation refers to the fact that the population has only been present at the HA site after a major eruption 350 years ago.

17. The TP53 identification looked interesting though I am not familiar with this analysis and I wondered about how one controls for spurious network patterns - some additional explanation would have been useful.

STRING-DB looks for connections between an inputted gene list using evidence from published experiments but can be set to also include evidence from additional published works. These can be refined through increasing stringency scoring. Genes with no evidence of connection can be hidden from the network.

18. Discussion
See above comments, this jumps too quickly into assumed adaptation and acclimation in the absence of fitness data and better replicated sampling without acknowledging the limitations of the current work.

We recognise the gold standard of research into adaption and acclimatisation uses fitness data of phenotypic response to backup the genetic picture identified. This limitation has been highlighted in the discussion (Lines 292)

July 20, 2022

RE: Life Science Alliance Manuscript #LSA-2022-01513-TR

Dr. Iain Perry
Cardiff University
Sir Geraint Evans Building
Heath Park
Cardiff CF14 4XN
United Kingdom

Dear Dr. Perry,

Thank you for submitting your revised manuscript entitled "Molecular insights into high altitude adaption and acclimatisation of *Aporrectodea caliginosa*". We would be happy to publish your paper in Life Science Alliance pending final revisions necessary to meet our formatting guidelines.

- please use the [10 author names, et al.] format in your references (i.e. limit the author names to the first 10)
- please add a callout for Figure 1A, Figure 3 B, C, D to your main manuscript text and add a figure legend section to the bottom of your manuscript, including the legend for the table
- you have a file labeled site data; is this table S1? if yes please relabel accordingly

A. FINAL FILES:

B. MANUSCRIPT ORGANIZATION AND FORMATTING:

Sincerely,

Reviewer #1 (Comments to the Authors (Required)):

This paper investigates genomic and transcriptomic differentiation (adaptation and acclimation) of earthworms of known colonisation history at high and low elevations, with environmental differences comparable in scale to projected climate change.

The work is technically sound and of sufficient scope to provide meaningful results. The nature of the problem addressed and the findings in relation to the original objectives address an important scientific challenge, which is to improve understanding of response to environmental variation in order to inform prediction of response to future climate change.

This is a revised version of the paper, and my analysis of points addressed in revision is that the current version of the paper is more balanced and sound. The main issue with the earlier version was lack of perspective on what is result, conclusion, and extrapolation. That issue has been addressed in the revision and I regard this paper as ready for publication.

Reviewer #2 (Comments to the Authors (Required)):

The authors have provided useful revisions to many of my main comments but also chosen to ignore some of these and I still think that the issues remain.

July 29, 2022

RE: Life Science Alliance Manuscript #LSA-2022-01513-TRR

Dr. Iain Perry
Cardiff University
Sir Geraint Evans Building
Heath Park
Cardiff CF14 4XN
United Kingdom

Dear Dr. Perry,

Thank you for submitting your Research Article entitled "Molecular insights into high altitude adaption and acclimatisation of *Aporrectodea caliginosa*". It is a pleasure to let you know that your manuscript is now accepted for publication in Life Science Alliance. Congratulations on this interesting work.

DISTRIBUTION OF MATERIALS:

Again, congratulations on a very nice paper. I hope you found the review process to be constructive and are pleased with how the manuscript was handled editorially. We look forward to future exciting submissions from your lab.

Sincerely,
